# Full-Thickness Tumor Resection of Oral Cancer Involving the Facial Skin—Microsurgical Reconstruction of Extensive Defects after Radical Treatment of Advanced Squamous Cell Carcinoma

**DOI:** 10.3390/cancers13092122

**Published:** 2021-04-28

**Authors:** Julius Moratin, Jan Mrosek, Dominik Horn, Karl Metzger, Oliver Ristow, Sven Zittel, Michael Engel, Kolja Freier, Juergen Hoffmann, Christian Freudlsperger

**Affiliations:** 1Department of Oral and Cranio-Maxillofacial Surgery, University of Heidelberg, Im Neuenheimer Feld 400, D-69120 Heidelberg, Germany; jan.mrosek@med.uni-heidelberg.de (J.M.); karl.metzger@med.uni-heidelberg.de (K.M.); oliver.ristow@med.uni-heidelberg.de (O.R.); sven.zittel@med.uni-heidelberg.de (S.Z.); michael.engel@med.uni-heidelberg.de (M.E.); juergen.hoffmann@med.uni-heidelberg.de (J.H.); christian.freudlsperger@med.uni-heidelberg.de (C.F.); 2Department of Oral and Cranio-Maxillofacial Surgery, Saarland University Hospital, Kirrberger Straße, D-66424 Homburg, Germany; dominik.horn@uks.eu (D.H.); kolja.freier@uks.eu (K.F.)

**Keywords:** oral cancer, free flaps, HNSCC

## Abstract

**Simple Summary:**

Advanced malignant tumors of the oral cavity are challenging because they impose serious oncological and functional requirements on the treatment specialist. Depending on the localization and the extent of the primary tumor, a full-thickness resection affecting the facial skin may be necessary to achieve a complete tumor resection. The resulting defects need adequate reconstruction in order to restore the aesthetics and functionality of the orofacial system. In this retrospective analysis, the authors aimed to evaluate treatment techniques for these tumors and analyze the clinical outcome of the related procedures. Full-thickness tumor resection with free flap reconstruction due to advanced cancer was performed in 33 patients.

**Abstract:**

Advanced tumors of the head and neck are challenging for the treatment specialist due to the need to synergize oncological and functional requirements. Free flap reconstruction has been established as the standard of care for defects following tumor resection. However, depending on the affected anatomic subsite, advanced tumors may impose specific difficulties regarding reconstruction, especially when full-thickness resection is required. This study aimed to evaluate reconstructive strategies and oncological outcomes in patients with full-thickness resection of the oral cavity. A total of 33 patients with extensive defects due to squamous cell carcinoma of the oral cavity were identified. Indications, reconstructive procedures, and clinical outcome were evaluated. Thirty-two patients (97%) presented locally advanced tumors (T3/T4). Complete tumor resection was achieved in 26 patients (78.8%). The anterolateral thigh flap was the most frequently used flap (47.1%), and the primary flap success rate was 84.8%. The cohort demonstrated a good local control rate and moderate overall and progression-free survival rates. Most patients regained full competence regarding oral alimentation and speech. Full-thickness tumor resections of the head and neck area may be necessary due to advanced tumors in critical anatomic areas. In many cases, radical surgical treatment leads to good oncological results. Free flap reconstruction has been shown to be a suitable option for extensive defects in aesthetically challenging regions.

## 1. Introduction

Head and neck squamous cell carcinomas (HNSCCs) are among the seven most common malignant tumors global and a significant portion of these tumors arises from the mucosa of the oral cavity [1]. Despite recent advances in the therapy of HNSCC by the introduction of immune checkpoint inhibition for recurrent/metastatic tumors, surgery and/or primary (chemo-)radiotherapy remain the only primary therapeutic options for most patients [2,3,4]. Advanced tumors, in particular, impose a challenge to both the patients suffering from the disease and the treating specialist, due to a trade-off between oncological necessities, and functional and aesthetic requirements, that is inevitable in many cases of head and neck tumor therapy. Although primary chemo-radiation is a relevant option for head and neck cancer patients, factors including tumor localization, prior adjuvant therapy in cases of recurrent tumors, other medical reasons, or the individual decisions of patients and treating specialists, leave surgery as the only curative therapy for advanced tumors in many cases.

The need for a complete resection of oral tumors with safe surgical margins may lead to extensive defects including full-thickness perforations involving the facial or the cervical skin. While small defects after surgical tumor ablation may be restored using local wound closure, split skin grafts, or regional flaps, extensive defects in most cases require the use of free flaps to achieve an adequate result in terms of function and aesthetics [5,6,7,8,9].

Although reconstructive surgery of the head and neck region is a well-established field in many centers, the preparedness and the expertise for extensive resections resulting in full-thickness defects and consequent reconstructions may vary. This may lead to undertreatment of patients, affecting both oncologic and functional outcomes in terms of post-therapeutic quality of life. The purpose of the presented study, therefore, was to present an overview of reconstructive strategies and clinical outcomes in a cohort of patients with full-thickness tumor resections of squamous cell carcinomas of the oral cavity.

## 2. Results

### 2.1. Patients

A total of 33 patients were included in the analysis; 12 (36.4%) were female and 21 (63.6%) were male, with a mean age of 67 ± 13.7 years and a range from 25 to 95 years. All patients received surgical therapy in the Department of Oral and Cranio-Maxillofacial Surgery of the Heidelberg University Hospital. Twenty-two (66.7%) patients suffered from primary oral squamous cell carcinoma and 11 (33.3%) were treated because of a local or locoregional tumor recurrence. Table 1 provides detailed information on demographic, clinical, and pathological features of the investigated cohort.

### 2.2. Therapeutic Procedures

All patients underwent surgical therapy due to primary or recurrent squamous cell carcinoma of the oral cavity. Complete tumor resection (R0) was achieved in 26 patients (78.8%), incomplete tumor resection (R1) was performed in six patients (18.2%). In one patient (3.0%) resection status was unclear (Rx) due to a fragmented preparation. Most cases of incomplete tumor resections (5/6, 83.3%) were related to tumors of the floor of the mouth/mandible.

The whole cohort of 33 patients had an indication for postoperative adjuvant therapy due to advanced tumor stage, the presence of neck node metastases, incomplete tumor resection status, histopathological risk factors (L+, V+, pN+), or a combination of several factors. In fact, only 15 patients (45.5%) received postoperative (chemo-)radiation. The remaining patients did not receive adjuvant treatment due to a history of prior radiotherapy (*n* = 11/33.3%), reduced medical condition or incomplete wound healing after surgery (*n* = 3/9.1%), and patients’ refusal (*n* = 4/12.1%).

### 2.3. Surgical Technique

All patients received full-thickness tumor resection due to an involvement of the external facial/cervical skin by the tumor mass and the intention to obtain tumor-free resection margins. The most commonly affected area was the cheek (*n* = 23/69.7%), followed by the submental/submandibular region (*n* = 10/30.3%), whereas the most commonly affected primary tumor sites were the mandible, including the floor of the mouth (*n* = 20/60.6%) and the buccal mucosa (*n* = 8/24.2%) (Table 1).

Free flap reconstruction was performed in all patients. Primary reconstruction was successful in 28 patients (84.8%) and flap loss occurred in five patients (15.2%). In all patients with a failure of the first flap, a second free flap reconstruction was performed successfully using an ALT flap (*n* = 5). The types of free flaps used for defect restoration included the anterolateral thigh flap (ALT, *n* = 24/82.7%), the radial forearm flap (RFF, *n* = 3/9.1%), the scapula flap (*n* = 3/9.1%), the fibula flap (*n* = 2/6.1%), and the latissimus dorsi flap (*n* = 1/3%). Soft-tissue flaps were prepared for the restoration of full-thickness defects by partial de-epithelization to create two separated skin paddles (Figure 1).

Table 2 and Table 3 provide an overview of types of free flaps used for reconstruction after full-thickness resection and corresponding success-rates and Table 4 provides a set of considerations regarding the ALT flap as technique of choice for full-thickness defects of the head and neck area.

### 2.4. Perioperative Management

The mean hospitalization period for all patients was 23.1 ± 10.9 days with a range from 9 to 53 days. Two patients (6.1%) died during their hospitalization period due to postoperative complications. A total of 24 patients (72.7%) were temporarily dependent on a tracheostomy for a mean duration of 23.4 ± 32.4 days. Twelve patients (36.4%) were temporarily or constantly dependent on a percutaneous endoscopic gastrostomy (PEG).

### 2.5. Outcome

#### 2.5.1. Oncological Outcome

The mean follow-up period was 21 ± 23 months with a range from 1 to 91 months. Overall survival, progression-free survival and the local control rate for the investigated cohort are depicted in Figure 2. Twelve patients (36.4%) experienced disease progression and 12 patients (36.4%) died during follow-up. The patients with disease progression developed a local recurrence in six cases (50%), regional recurrence in two cases (16.7%), and distant metastases in four cases (33.3%), all of which were located in the lung.

#### 2.5.2. Functional Outcome

In all patients, adequate reconstruction was achieved using a free flap. Although 12 patients (36.4%) were temporarily dependent on a percutaneous gastrostomy, the remaining patients (*n* = 21/63.6%) were able to maintain sufficient oral nutrition postoperatively. Figure 2 exemplifies the preoperative situation, surgical procedures, and postoperative outcome of a patient with recurrent squamous cell carcinoma of the maxilla. Figure 3 and Figure 4 exemplify the preoperative situation, and postoperative outcome of two patients with advanced squamous cell carcinoma of the oral cavity.

## 3. Discussion

Despite advances in screening and diagnosis of head and neck cancer, a significant proportion of patients presents with locally advanced disease [10,11]. As surgical therapy remains the primary modality for the treatment of locally advanced oral squamous cell carcinomas (OSCCs), the extent of tumor resection has to be carefully balanced against possible functional and aesthetic impairments affecting the patient’s quality of life. In patients in which oral cavity tumors exhibit involvement of the external skin, in particular, extensive resection might be challenging and usually requires the use of free flaps for reconstruction. As residual disease is accompanied by worse overall and progression-free survival, a critical goal of surgery must be the achievement of clear oncologic margins [12]. Nonetheless, radical surgical intervention may lead to functional and aesthetic impairments.

Hence, the present manuscript exclusively focuses on patients with locally advanced OSCCs involving the external skin, independent of their medical history regarding the sequence of the disease. This heterogeneous cohort integrating patients with primary and recurrent disease was chosen because it offers the possibility to exemplify and discuss aspects of synergism between surgical oncology and reconstructive surgery.

The main objective of oncological surgery is to create tumor-free margins, which in this cohort was successfully achieved in 78.8% of the cases. Although this rate generally might be critically discussed, it has to be considered under the aspect of the dominance of advanced tumors in this cohort (97% T3/T4 tumors), including 36.4% recurrent tumors and a history of prior multi-modality-treatment in 33.3% of the patients. Therapeutic considerations for patients with primary disease are generally different from those with recurrent disease, especially regarding potential alternatives. However, the presented cohort offers certain peculiarities that should be respected concerning treatment planning. Although primary radiotherapy is a valuable option for radiation-naive patients suffering from head and neck cancer that offers good local and regional control rates, there are advantages of a primary surgical approach that should be considered [13]. These include the possibility to further improve the outcome if complete tumor resection is achieved, to guide adjuvant therapy by possibly detecting occult lymph node metastases, and to restore areas where extensive tumor growth may lead to defect healing if primary radiotherapy is applied. As other therapeutic approaches including the newly introduced immune-checkpoint inhibition are, at present, restricted to palliative settings and therapeutic response is limited to a fraction of patients, the affected patients need profound guidance regarding the implications of the proposed therapies and their rejection [4,14].

Several retrospective and prospective studies evaluated the concept of salvage surgery for patients suffering from recurrent head and neck cancer, mostly advocating the use of free flaps for the reconstruction of large tissue defects [14,15,16,17,18]. Although the term “salvage surgery” mostly applies to tumor resections with curative intent in patients with recurrent tumors and a history of previous (radio-) therapies, and consequently lacking alternatives to the surgical approach, the focus of the present study lies on extensive resections of head and neck tumors irrespective of the sequence of prior therapy to exemplify the possibilities and limitations of tumor surgery of the head and neck in general. Consequently, the presented cohort is heterogeneous regarding the medical history of the patients and limited in size due to the focus on cases with a need to perform full-thickness tumor resections. Although most of the mentioned publications focus on salvage surgery approaches, the general considerations mostly also apply to patients with advanced primary tumors in sensitive localizations. In every case, critical balancing of the patient’s individual situation and wishes is urgently warranted, including prognosis, the patient’s physical and mental capacity, and the chance for therapeutic success.

The oncological results in our analysis demonstrate a high local control rate and, although the overall and progression-free survival rate is typical for a cohort of patients suffering from advanced head and neck cancer, only six cases (18.2%) of local recurrence occurred, and the remaining six cases of disease recurrence were attributable to regional or distant metastases. These results, although drawn from a small cohort, are comparable to those reported in the available literature and confirm the validity of our data [19,20]. Furthermore, in many of the presented cases, surgery was the only available curative therapeutic option due to a history of radiotherapy in the cases of recurrent tumors. In other cases, a definitive radio(-chemo)-therapy was abandoned because of the tumor’s vicinity or its manifest infiltration of sensitive structures with a high risk of postoperative osteonecrosis (e.g., the mandible) [21]. Additionally, extensive tumors carry a risk of a defect healing after radiotherapy, especially if they cause a connection between the oral cavity and the facial skin. In those cases, a primary surgical approach to restore the anatomic integrity of the oral cavity and the facial skin appears sensible for obvious aesthetic and functional reasons, because there is plethora of publications on the higher complication rate of reconstructive surgical approaches in patients with a history of radiotherapy [22,23,24].

The ALT was the most frequently used flap in our cohort. It is a versatile option for the extensive defects due to its volume, good pedicle length, and comparably low donor site morbidity [25,26,27,28]. It is especially suitable for full-thickness defects, because partial de-epithelization of the flap allows for the creation of separate skin paddles that may be tunneled through the defect and may be used for the restoration of the intra- and extraoral parts of the defect even if only one perforator is present. Other flaps used in our cohort were the radial forearm flap, the scapula flap, the fibula flap, and the latissimus dorsi flap. The overall flap success rate in our cohort was 84.8%, which is inferior to that reported by other authors [18,24]. Flap success rates were different between patients with primary (flap success: 90.5%) and recurrent disease (75%), which may be explained by the oft-reported tendency towards higher complication rates in patients with a history of prior reconstructive approaches and radiotherapy [24,29,30]. Additionally, the general need for bigger flaps for full-thickness defects may carry an increased risk of flap failure.

Several authors have reported on techniques for the restoration of facial full-thickness defects. They proposed free flaps including the ALT, the fibula flap, and the radial forearm flap, alone or in combination with rotational flaps for the extraoral parts of the defect [31,32,33,34]. As most of the reported studies focused on small numbers of patients, to the best of our knowledge this article describes one of the largest cohorts with full-thickness defects.

Although there was an accumulation of tumors in distinct subsites of the oral cavity (mandible/floor of the mouth, buccal mucosa, maxilla) eventually leading to full-thickness tumor resection, each patient needs individual treatment planning and, thus, patients with similar expected defects may require different reconstructive approaches to achieve an optimal or at least satisfying outcome. Several factors are of critical relevance for the choice of the reconstructive technique, including medical history, the sequence of reconstruction, and the patient’s individual physical condition.

This study has several limitations. The retrospective nature of the analysis is problematic regarding the case selection and the significance of deduced statements. As stated previously, the focus on a special subset of tumor patients that was mainly defined by the necessity for full-thickness resection led to a small cohort. The inclusion of primary and recurrent tumors aggravates the interpretation of the survival data and does not allow for valid conclusions. Nevertheless, the authors think that the presented study may be regarded as a legitimate suggestion for a possible treatment regime for a subset of head and neck cancer patients that is relevant and challenging regarding oncological and reconstructive considerations.

## 4. Materials and Methods

The presented retrospective study was conducted with respect to the principles of the Declaration of Helsinki and was approved by the local ethics committee of the University of Heidelberg (Ethic vote: S-183/2015). Written informed consent was given by all patients. All patients receiving full-thickness tumor resection of a squamous cell carcinoma of the oral cavity (primary and recurrent tumors) in the Department of Oral and Cranio-Maxillofacial Surgery of the Heidelberg University Hospital between 2010 and 2020 were included in the analysis. Secondly, patients who received full-thickness tumor resections were identified for further investigation. Demographic, clinical, and pathological data were extracted from the electronic patient records and transferred to an anonymized data processing sheet.

Statistical analysis was performed using IBM SPSS Statistics^®^ 25 (IBM, Armonk, New York, NY, USA). Categorial data were described using descriptive analysis.

Data regarding overall and progression-free survival data and follow-up recalls were collected from the electronic patient document and transferred in SPSS. Survival was assessed using the Kaplan–Meier method as the time between initial therapy and the date of death or last follow-up (censored) for overall survival, and between initial therapy and the date of local, regional or distant disease progression or last follow-up (censored) for progression-free survival and local control rate, accordingly. Data cutoff refers to 31st December 2020.

## 5. Conclusions

Surgical tumor therapy with curative intent is based on the achievement of tumor-free margins wherever possible. If clear margins seem achievable preoperatively, the surgical procedures including reconstructive strategies should integrate oncological and functional considerations to obtain an optimal outcome in terms of survival and postoperative quality of life. Extensive tumor resections in the head and neck area including full-thickness resections are feasible, and free flap reconstruction offers a versatile tool to restore function and aesthetics while providing good local disease control rates.

## Figures and Tables

**Figure 1 cancers-13-02122-f001:**
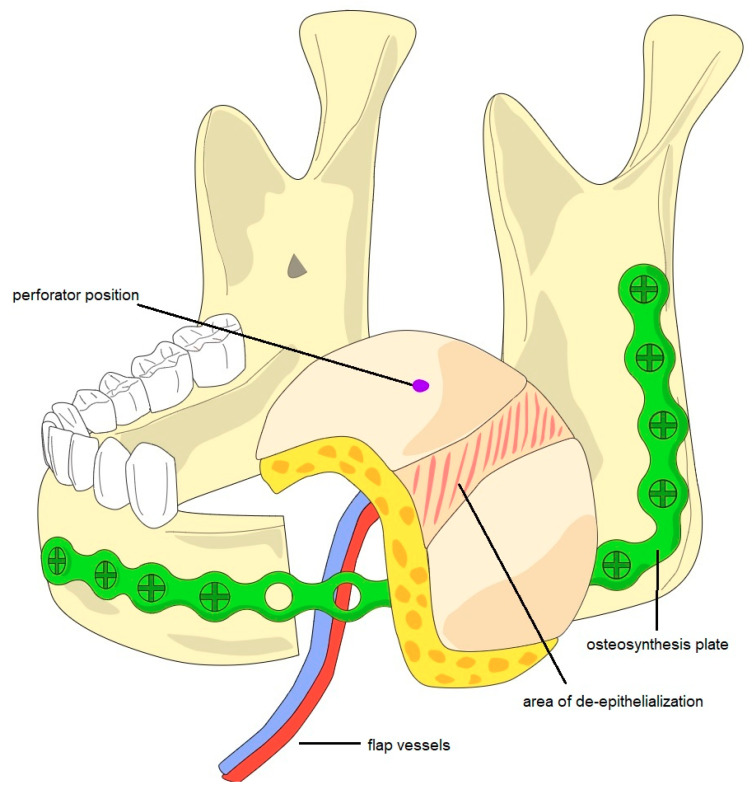
Schematic diagram of a partially de-epithelialized ALT flap and an osteosynthesis plate for the reconstruction of a full-thickness defect after tumor resection with segmental resection of the left-sided mandible. After segmental resection and the individualization of an osteosynthesis plate, the ALT flap was harvested. The prepared flap then underwent partial de-epithelialization in order to create two skin paddles for restoration of the oral and the facial defect.

**Figure 2 cancers-13-02122-f002:**
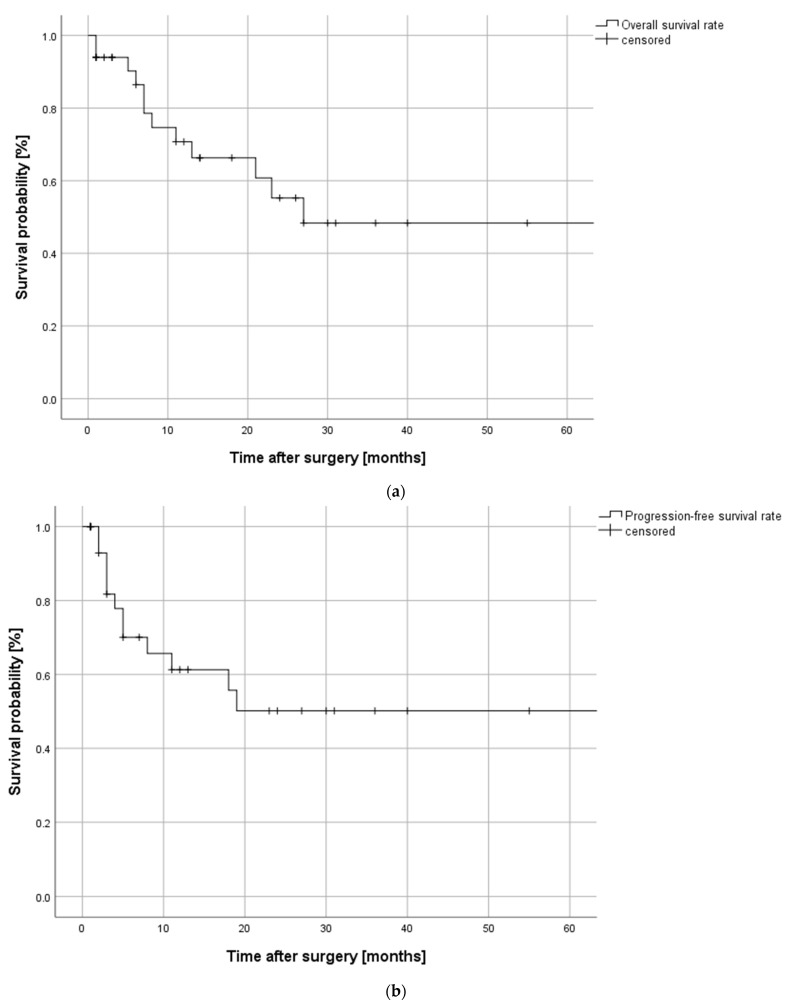
Kaplan–Meier curves depicting (**a**) overall survival, (**b**) progression-free survival, and (**c**) local control rate of the investigated cohort with full-thickness resections following advanced oral squamous cell carcinoma during a follow-up period of up to 60 months. Overall and progression-free survival rates exhibit a decrease to about 50% after 20–30 months, the local control rate shows a decrease to about 75% after 20 months.

**Figure 3 cancers-13-02122-f003:**
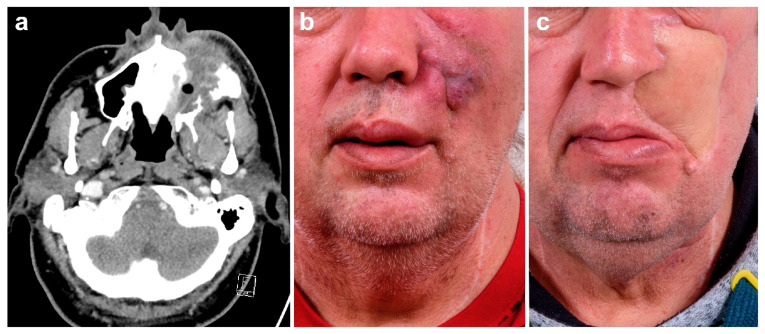
Male patient with recurrent squamous cell carcinoma of the left-sided maxilla with infiltration of the maxillary sinus and facial skin. (**a**) CT scan depicting the infiltration of the paranasal sinus and the facial skin. (**b**) Full-frontal image of the preoperative situation with tumor progression of the left cheek. (**c**) Full-frontal image of the postoperative situation after full-thickness tumor resection and reconstruction with ALT flap.

**Figure 4 cancers-13-02122-f004:**
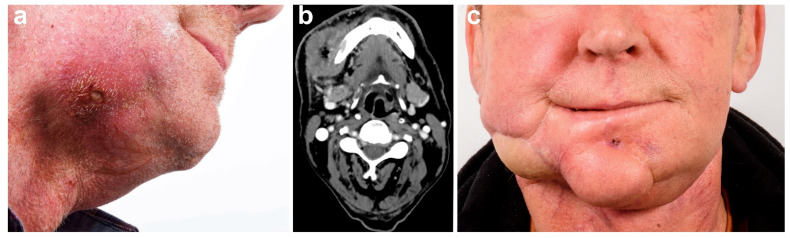
Male patient with primary squamous cell carcinoma of the right-sided mandible with infiltration of the facial skin. (**a**) Tumor infiltration of the right cheek. (**b**) CT scan depicting the tumor of the right-sided mandible with infiltration of the facial skin. (**c**) Full-frontal image of the postoperative situation after full-thickness tumor resection and reconstruction with ALT flap.

**Table 1 cancers-13-02122-t001:** Descriptive data regarding demographic and clinical features of the investigated cohort.

Parameter	Number of Cases (%)
Gender	
Female	12 (36.4)
Male	21 (63.6)
Age	
	Mean: 67 ± 13.7 years
	Range: 25–95 years
Risk Factors	
Tobacco	17 (51.5)
Alcohol	10 (30.3)
Primary or Recurrent Tumor	
Primary tumor	21 (63.6)
Recurrent tumor (local recurrence)	12 (36.4)
Tumor Localization	
Mandible + Floor of the mouth	20 (60.6)
Buccal mucosa	8 (24.2)
Maxilla	5 (15.2)
T Stage	
T1	-
T2	1 (3.0)
T3	6 (18.2)
T4	26 (78.8)
N Stage	
0	21 (63.6)
1	1 (3.0)
2a	-
2b	1 (3.0)
2c	2 (6.1)
3	8 (24.2)
M Stage	
0	33 (100)
1	-
UICC Stage	
1	-
2	-
3	4 (12.1)
4	29 (87.9)
Histopathological Tumor Features	
Lymphatic invasion	15 (45.5)
Vascular invasion	4 (12.1)
Perineural invasion	12 (36.4)

**Table 2 cancers-13-02122-t002:** Overview of reconstructive strategies and success rates in relation to affected anatomical subsite of the oral cavity.

Affected Subsite of the Oral Cavity	Free Flap (no.)	Success Rate
Mandible + Floor of the Mouth		
	ALT (*n* = 15)	12/15 (80%)
	RFF (*n* = 1)	1/1 (100%)
	Scapula (*n* = 1)	1/1 (100%)
	Fibula (*n* = 2)	2/2 (100%)
Buccal Mucosa		
	ALT (*n* = 7)	7/7 (100%)
	RFF (*n* = 1)	1/1 (100%)
Maxilla		
	ALT (*n* = 2)	2/2 (100%)
	RFF (*n* = 1)	1/1 (100%)
	Scapula (*n* = 2)	0/2 (0%)

**Table 3 cancers-13-02122-t003:** Overview on free flaps used for the restoration of full-thickness defects with revision and success rates.

Type of Free Flap	Revision Rate	Success Rate
ALT	4/24 (16.7%)	21/24 (87.5%)
RFF	0/3	3/3 (100%)
Scapula	3/3 (100%)	1/3 (33.3%)
Fibula	0/2	2/2 (100%)
Latissimus dorsi	0/1	1/1 (100%)

**Table 4 cancers-13-02122-t004:** Considerations regarding the ALT flap as reconstructive technique of choice for full-thickness defects of the head and neck area.

**Advantages of ALT Flaps**
Good pedicle length
Especially suitable for extensive defects due to its bulkiness
Partial de-epithelization allows for the restoration of several neighboring defects, even if only one perforator is found
Low donor-site morbidity and high rate of primary closure of donor-sites
Low surgery times due to the possibility of a two-team approach
**Disadvantages of ALT Flaps**
Risk of aesthetic and functional impairment if a bulky flap is used for narrow defects
Risk of compromising the perforator if ALT is used for the coverage of metal plates
Need for secondary osseous reconstruction in cases of mandibular resections

## Data Availability

The data presented in this study are available on request from the corresponding author.

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
