# Peer review of "Full-Thickness Tumor Resection of Oral Cancer Involving the Facial Skin—Microsurgical Reconstruction of Extensive Defects after Radical Treatment of Advanced Squamous Cell Carcinoma"

_cancers, 2021, doi:10.3390/cancers13092122_

Round 1

Reviewer 1 Report

Cancers 21. 03. 21

Manuscript is not corrected as recommended (see previous recommendation of 17.02.21)

Especially Figure 1 and 2  Authors  should correct figures as recommended

Authors added new figure Fig 3. Authors need to exclude Fig3 CDF, show only  ab  and  e-only with region after surgery no face of the patient even if permission from patient is received in written form.  Show drawing for ALT flaps for this surgery in details same as recommended for figure 2 in previous review

Cancers 17.02.21

Abstract should be concise and  clear with clear conclusion  of the recommended method

It is not clear is this prospective or retrospective study

Authors need to elaborate more on neck squamous cell carcinoma properties in the section Introduction and discussion including clinical feature. Also, Authors need elaborate more on chemotherapy and radiotherapy of this cancer and show correlation with surgery as a last option in the caner treatment, because surgery is not treatment of choice for cancers.

Figure 1 Legend need to be explained in details Kaplan-Meier curves for Overall survival, Progression-free survival

Figure 2 needs to be rearranged to show CT then drawing for ALT flap (in details)  rather then person surgery even identity of the person. Just show Region of the operation rather than all face of the person. Exclude Fig2cd and and Fig2 be shown only region  before and after surgery and CT

In The Discussion explain in details why ALT is the best method  instead radiotherapy and chemotherapy and why surgery has supeioritry over radiotherapy and chemotherapy

Author Response

As suggested, we included a drawing for an ALT flap used for the reconstruction after a full-thickness resection (new Figure 1). The legend for the Kaplan-Meier curves (Figure 2) were enhanced. Concerning the figures 3 and 4 (patient cases) the authors discussed the recommendation of the reviewer. Still, the majority of the authors think that the figures in their present form best exemplify the purpose of this article, especially regarding the positive comments of the other two reviewers. We would like to leave the decision regarding the final form of the figures to the editor. We will gladly remove or revise the patient cases if the editor feels that this is necessary to improve the quality of the article.

Reviewer 2 Report

Extensive syntax and grammar editing should be accomplished. I suggest a careful revision of the entire manuscript to improve readability.

Patients and Methods

-Patients and methods section should follow introduction.

-Line 291 “perforating tumor resections” should be revised.  Do you mean full-thickness? Please clarify and revise throughout the entire manuscript.

-Line 98-100 “Survival was assessed as the time between initial therapy 298 and the date of death or last follow-up (censored data) for overall survival, and between 299 initial therapy and the date of local, regional or distant disease progression or last follow-300 up (censored data) for progression-free survival, accordingly.”

What is a censored data?

-line 295 the statistical analysis should be described in detail

Author Response

Comments:

Extensive syntax and grammar editing should be accomplished. I suggest a careful revision of the entire manuscript to improve readability.

Answer:
As suggested, we sent the manuscript to the MDPI English Editing Service. Please find enclosed the fully revised manuscript after English Language Editing including the respective certificate.

Patients and Methods

-Patients and methods section should follow introduction.

Answer:
We used a template for the preparation of this manuscript that was provided by the journal. The sequence of the headings and sub-headings is predetermined by this template and corresponds to other “cancers” articles that have been published.

-Line 291 “perforating tumor resections” should be revised. Do you mean full-thickness? Please clarify and revise throughout the entire manuscript.

Answer:
We want to apologize for this repetitive flaw from the first draft the manuscript. We changed the term to “full-thickness tumor resection” as suggested by the reviewer.

-Line 98-100 “Survival was assessed as the time between initial therapy 298 and the date of death or last follow-up (censored data) for overall survival, and between 299 initial therapy and the date of local, regional or distant disease progression or last follow-300 up (censored data) for progression-free survival, accordingly.”

What is a censored data?

Answer:
We changed the term “censored data” to “censored”. This term characterizes time points to which patients are excluded from the survival analysis when the last follow-up has been reached. To the best of our knowledge, this terms are appropriate and conventional in survival time analysis.

-line 295 the statistical analysis should be described in detail

Answer:

As suggested, we enhanced the M&M section to provide more details on the statistical analysis. Here, we included more information on the electronic patient documents, the end of data collection and the details of the survival analysis.

Reviewer 3 Report

The article improved a lot compared to the previous submission. The paper is in my opinion publishable.

Author Response

Comment:

The article improved a lot compared to the previous submission. The paper is in my opinion publishable.

Answer:

We thank the reviewer for this comment and hope to be able to persuade the remaining reviewers and editor to accept our manuscript.

Round 2

Reviewer 1 Report

Authors should change the figures as recommended  previously 

Fig 1 ALT Flap need details through all steps

Fig 3 show  a),   b)  only bellow eyes  and e) only bellow eyes,  rest  exclude 

Fig 4 Show  a, b) only bellow eyes  and e) only bellow eyes, rest  exclude 

Author Response

Dear Reviewer,

we modified figures 3 and 4 as suggested by the reviewer. Furthermore, we added captions to Figure 1 and enhanced the figure legend in order to describe the procedures in detail. The flap harvesting was not described, as there are numerous publications by other authors on ALT harvesting.

Reviewer 2 Report

In table 1, data about age should be written in the right column

Kaplan Meier estimator should be mentioned in MM section, Statistical analysis paragraph

line 305 the words "perforating tumor resection" should be abandoned as previously stated. (1st round of review)

Author Response

Dear Reviewer,

the data about age were moved to the right column of table 1 as suggested.

The M&M section was modified in order to mention the Kaplan-Meier method (“Survival was assessed using the Kaplan-Meier method as the time between initial therapy and the date of death or last follow-up (censored) for overall survival, and between initial therapy and the date of local, regional or distant disease progression or last follow-up (censored) for progression-free survival and local control rate, accordingly.”

We changed the term “perforating tumor resection” to “full-thickness tumor resection” and apologize for this repetitive flaw.

This manuscript is a resubmission of an earlier submission. The following is a list of the peer review reports and author responses from that submission.

Round 1

Reviewer 1 Report

Abstract should be concise and  clear with clear conclusion  of the recommended method

It is not clear is this prospective or retrospective study

Authors need to elaborate more on neck squamous cell carcinoma properties in the section Introduction and discussion including clinical feature. Also, Authors need elaborate more on chemotherapy and radiotherapy of this cancer and show correlation with surgery as a last option in the caner treatment, because surgery is not treatment of choice for cancers.

Figure 1 Legend need to be explained in details Kaplan-Meier curves for Overall survival, Progression-free survival

Figure 2 needs to be  rearranged to show CT then drawing for ALT flap (in details)  rather then person surgery even identity of the person. Just show Region of the operation rather than all face of the person. Exclude Fig2cd and  and Fig2 be show only region  before and after

In The Discussion explain in details why ALT is the best method  instead radiotherapy and chemotherapy and why surgery has supeioritry over radiotherapy and chemotherapy

Reviewer 2 Report

The authors described a retrospective study on 33 patients underoing advanced SCC treatment. The study population is tough to manage, and good oncologic and functional results are not always satisfactory.

The structure of the study should be improved; the Discussion is not well focused on the purpose of the study. However, the oncologic results are satisfactory and the presented clinical case is good.

Extensive syntax and grammar editing should be accomplished. I suggest a careful revision of the entire manuscript to improve readability.

 Furhtermore, the following issues should be addressed with a reply point-by-point

Title

“Perforate” and “tumor” are more often described as “perforation of an organ due to a tumor”. I suppose that you mean perforating as “full-thickness” tumor resection.

Finally, I suggest to revise the title (and throughout the entire manuscript) because “Perforating tumor resection” sounds awkward.

The title should include also the reconstructive stage of the treatment. I suggest that you add “microsurgical reconstruction” or “free flap reconstruction” or something similar to the title. There could not be any good functional outcome without reconstruction.

Introduction:

Page 2 line 54-57. The bibliography in this paragraph is not well updated, articles from ‘90…. I would suggest replace at least one or two of the references 5,6,7 and add a more updated bibliography. I could suggest among others:

Di Taranto G, et al. Outcomes following head neck free flap reconstruction requiring interposition vein graft or vascular bridge flap. Head Neck. 2019 Sep;41(9):2914-2920

Podrecca S, et al. Review of 346 patients with free-flap reconstruction following head and neck surgery for neoplasm. J Plast Reconstr Aesthet Surg. 2006;59(2):122-9.

Hurvitz KA, et al. Current options in head and neck reconstruction. Plast Reconstr Surg. 2006 Oct;118(5):122e-133e.  Line 59. Please revise and replace the word “preparedness”

Patients and Methods

Please revise patients and methods and Results sections. they are misplaced

Table 1 should report Mean age and SD as mentioned at page 2 line 69

Line 133. figure 3 was not provided by the authors

I would suggest to add a subheading named “Surgical technique” after patients and methods section to enhance your manuscript. You could describe some keypoint of your oncologic resection (if any) and how you provide double layer reconstruction with alt flap (two perforators?) and radial flap, and whatever tips and tricks for the reconstructive surgeons. The cases in your cohort are very tough, and the good result that your reached is not an easy target: this should be clear to the readers. Furthermore, after reading the last paragraph of introduction, the reader will expect some pearls and pitfalls.

Results

Authors should provide data about complication rate and build a table with complications and flap revision rate; which flaps failed and which flaps were harvested after the failures etc…

Line 134 “aesthetic outcome….” How did you evaluate it?

Figure 2. The clinical case shows a very good result. Please add the pre-post pictures of another case. Furthermore a picture with mouth closed and open would be perfect to show functional result and would appreciated by the readers

Discussion

Please provide discussion about your free flap survival rate. Do you think that double layer reconstruction is affecting your outcomes? Support with references

The discussion needs to include more data from the existing knowledge about full-thickness combined reconstruction of the oral cavity and skin

Reviewer 3 Report

The article is a retrospective study on all the patients with perforating tumor resection due to squamous cell carcinoma of the oral cavity treated in a single center in the period between 2010 and 2020. Although the article is interesting, as it is a very well documented case series, there are a lot of similar articles in the literature.  Also, no statistical analysis was basically performed, as the number of patients is not very high, and this is  a major limitation of the study. In my opinion, the article may be more well suited for a lower impact journal.